# Large Language Models Can Self-Improve

**Jiaxin Huang**[1*]  **Shixiang Shane Gu**[2]  **Le Hou**[2†]  **Yuexin Wu**[2]  **Xuezhi Wang**[2]
**Hongkun Yu**[2]  **Jiawei Han**[1]
[1]University of Illinois at Urbana-Champaign    [2]Google
[1]{jiaxinh3, hanj}@illinois.edu   [2]{shanegu, lehou, crickwu,
xuezhiw, hongkuny}@google.com

## Abstract

Large Language Models (LLMs) have achieved excellent performances in various tasks. However, fine-tuning an LLM requires extensive supervision. Human, on the other hand, may improve their reasoning abilities by self-thinking without external inputs. In this work, we demonstrate that an LLM is also capable of self-improving with only unlabeled datasets. We use a pre-trained LLM to generate "high-confidence" rationale-augmented answers for unlabeled questions using Chain-of-Though (CoT) prompting and self-consistency, and fine-tune the LLM using those self-generated solutions as target outputs. We show that without any ground truth label, our approach significantly improves the general reasoning ability of PaLM 540B model (74.4%→82.1% on GSM8K, 90.0%→94.4% on OpenBookQA, and 63.4%→67.9% on ANLI-A3) and can also be adapted to extreme low-resource cases where even training questions and CoT prompts are limited. We conduct ablation studies and show that fine-tuning on diverse reasoning paths is critical for self-improvement.

## 1 Introduction

Scaling has enabled Large Language Models (LLMs) to achieve state-of-the-art performance on a range of Natural Language Processing (NLP) tasks (Wang et al., 2018, 2019; Rajpurkar et al., 2016). More importantly, new capabilities have emerged from LLMs as they are scaled to hundreds of billions of parameters (Wei et al., 2022b): in-context few-shot learning (Brown et al., 2020) makes it possible for an LLM to perform well on a task it never trained on with only a handful of examples; Chain-of-Thought (CoT) prompting (Wei et al., 2022c; Kojima et al., 2022) demonstrates strong reasoning ability of LLMs across diverse tasks with or without few-shot examples;

---

[*]Work was done during Google internship.
[†]Corresponding author.

self-consistency (Wang et al., 2022c) further improves the performance via self-evaluating multiple reasoning paths.

Despite these incredible capabilities of models trained on large text corpus (Brown et al., 2020; Chowdhery et al., 2022), fundamentally improving the model performances beyond few-shot baselines still requires finetuning on an extensive amount of *high-quality supervised* datasets. FLAN (Wei et al., 2021; Chung et al., 2022) and T0 (Sanh et al., 2022) curated tens of benchmark NLP datasets to boost zero-shot task performances on unseen tasks; InstructGPT (Ouyang et al., 2022) crowd-sourced many human answers for diverse sets of text instructions to better align their model to human instructions; Minerva (Lewkowycz et al., 2022) parsed the full ArXiv database carefully for relevant articles to excel on challenging competitive math and science datasets. The need for large annotated data for supervised LLM training still remains a burden for low-resource applications or specific domains where only limited annotations are available.

In this paper, we study how an LLM capable of in-context few-shot learning and chain-of-thought reasoning, is able to *self-improve* its reasoning ability without supervised data. We show that using only input sequences (without ground truth output sequences) from multiple NLP task datasets, a pre-trained LLM is able to improve performances for both in-domain and out-of-domain tasks. Our method is shown in Figure 1: we first sample multiple predictions using few-shot Chain-of-Thought (CoT) (Wei et al., 2022c) as prompts, filter "high-confidence" predictions using majority voting (Wang et al., 2022c), and finally finetune the LLM on these high-confidence predictions. The resulting model shows improved reasoning in both greedy and multi-path evaluations. We call the model fine-tuned in this way as **Language Model Self-Improved (LMSI)**.

Note that LMSI depends on in-context few-shot learning and chain-of-thought reasoning abilities which small language models do not necessarily have. We empirically verify **LMSI** using a pre-trained 540B PaLM model (Chowdhery et al., 2022), where our method not only significantly improves training task performances (74.4%→82.1% on GSM8K, 90.0%→94.4% on OpenBookQA, and 63.4%→67.9% on ANLI-A3), but also enhances out-of-domain (OOD) tasks, without relying on supervised ground truth answers. Lastly, we explore more extreme cases where training questions and human-curated CoTs are also limited, and propose self-generating additional input questions and few-shot CoT prompts for model self-improving. We hope our simple approaches and strong empirical results could inspire more future work by the community to investigate optimal performances of pretrained LLMs without additional human supervision.

Our contributions are summarized as follows:

- We demonstrate that a large language model can self-improve by taking datasets without ground truth outputs, by leveraging CoT reasoning (Wei et al., 2022c) and self-consistency (Wang et al., 2022c) to generate diverse reasoning paths for self-training, and can achieve great improvments on in-domain multi-task performances as well as out-of-domain generalization.

- We provide detailed ablation studies on training sample formatting and sampling temperature after fine-tuning, and identify critical design choices for most successful self-improvement by LLMs.

- We further propose two approaches for model self-improving under extreme low-resource cases where even training questions and CoT prompts are limited, and achieve 74.2% on *zero-shot* GSM8K, against 43.0% by Kojima et al. (2022) or 70.1% through its naive extension with Wang et al. (2022c).

The rest of this paper is organized as follows. Section 2 discusses related work. Section 3 lays out our method in detail. Section 4 shows our setup for experiments. Section 5 demonstrates our experiment results with ablation studies. Section 6 concludes our work. The chain-of-thought prompts used in our work are included in Appendix A.

## 2 Related Work

**Learning from explanations.** Augmenting a machine learning model with explanations has been studied in existing literature extensively. For example, in the supervised learning setting, a model can be fine-tuned using human-annotated rationales (Zaidan et al., 2007; Ling et al., 2017a; Narang et al., 2020; Camburu et al., 2018; Cobbe et al., 2021; Chung et al., 2022). A few works have also looked at how explanations can help the models in various settings, e.g., in-context learning (Lampinen et al., 2022) and in distillation (Pruthi et al., 2022). Lightman et al. (2023) treat explanations as process supervision to train a reward model. In this paper, we focus more on the *unsupervised learning* setting, where we do not assume we have a rationale-augmented training dataset available, since human-annotated rationales can be expensive.

**Few-shot explanations improves reasoning in LLMs.** Recently, a lot of progress has been made towards improving LLMs' reasoning abilities via prompting or in-context learning. Wei et al. (2022c) propose Chain-of-Thought prompting, which prompts the language model to generate a series of natural-language-based intermediate steps, and show it can help language models better solve complex and multi-step reasoning tasks, with recent study (Wang et al., 2022a) analyzing the relevant contents and correct reasoning order being the most crucial factor of the success of Chain-of-Thought prompting. Wang et al. (2022c) improve Chain-of-Thought prompting by sampling multiple diverse reasoning paths and finding the most consistent answers via majority voting. Kojima et al. (2022); Zhang et al. (2022) propose to prompt the language model with "Let's think step by step" to generate reasoning in a zero-shot fashion. Zhou et al. (2022) decompose the questions into multiple sub-questions, and ask the language model to solve each sub-question sequentially.

**Refining explanations.** More recent work proposes to further refine the generated reasoning paths as some of them could be unreliable. For example, Ye and Durrett (2022) calibrate model predictions based on the reliability of the explanations, Jung et al. (2022) show that inducing a tree of explanations and inferring the satisfiability of each explanation can further help judge the correctness of explanations. Li et al. (2022a) show that

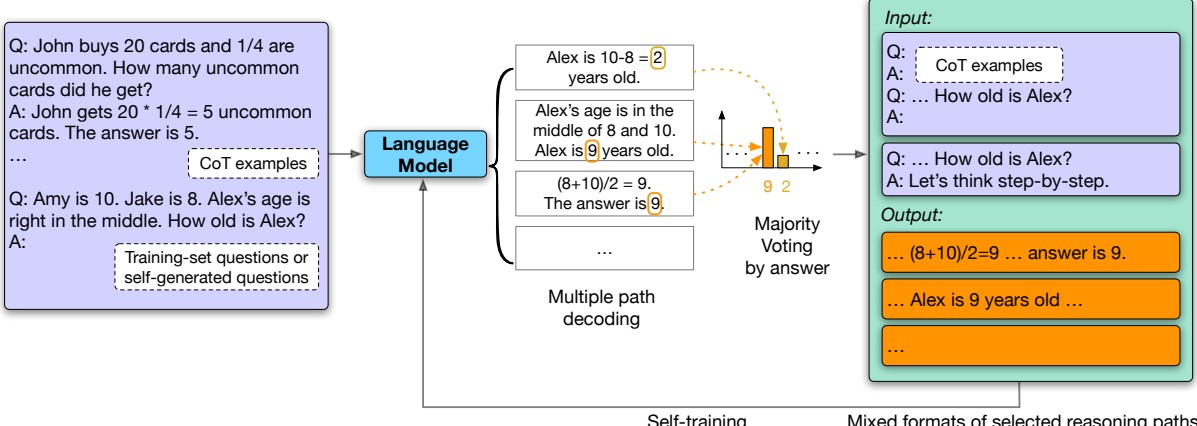

Figure 1: Overview of our method. With Chain-of-Thought (CoT) examples as demonstration (Wei et al., 2022c), the language model generates multiple CoT reasoning paths and answers (temperature $T > 0$) for each question. The most consistent answer is selected by majority voting (Wang et al., 2022c). The CoT reasoning paths that lead to the answer with the highest confidence are augmented by mixed formats, and are fed back to the model as the final training samples.

sampling a diverse set of prompts from the training data, and a voting verifier can be used to improve model's reasoning performance. Xi et al. (2023) and Zheng et al. (2023) propose to polish the problem progressively before the model reaching a stable answer. Zelikman et al. (2022) proposes better rationale generation by augmenting ground truth answers as hints when predicted answers are incorrect. Our work is orthogonal to these lines of work, as we utilize refined explanations for model self-improvement, and could readily incorporate these other refinement techniques for generating higher-quality self-training data. Our work is closely related to Zelikman et al. (2022) where we both propose to fine-tune a model on self-generated CoT data, but our method does not require ground truth labels and shows stronger empirical results with multi-task generalization. Different from existing work, we show that a mixture of the reasoning-path refinement techniques can be combined to further improve the quality of the generated reasoning paths, which is shown to be effective in boosting model's performance via self-improvement.

**Self-training models.** One related line of work is self-training (see a survey from Amini et al. (2022)). The key idea is to assign pseudo labels from a learned classifier to unlabeled data, and use these pseudo-labeled examples to further improve the original model training, e.g., (RoyChowdhury et al., 2019; Xie et al., 2020; He et al., 2020; Chen et al., 2021). Different from such prior work, our proposed self-improvement framework uses CoT prompting plus self-consistency to obtain high-confidence solutions on a large set of unlabeled data to augment the fine-tuning process.

**Distillation and dark knowledge.** Language models are known to preserve parametric knowledge (Schick and Schütze, 2020a,b) during the pre-training stage. Our method tangentially relates to rich literature on distillation (Ba and Caruana, 2014; Hinton et al., 2015), where a student network imitates a teacher network's classifier predictions on input examples. A key detail is to learn from soft targets instead of hard predicted labels, as softmax outputs with a high temperature reveal more detailed relative class likelihoods, colloquially known as *dark knowledge* (Hinton et al., 2015; Korattikara Balan et al., 2015). Recent studies (Zelikman et al., 2022; Snell et al., 2022; Eisenstein et al., 2022) show that *dark knowledge* within LLMs can be retrieved with more computation at inference time, such as adding informative instructions into the input sequence and output CoT generation (Wei et al., 2022c; Kojima et al., 2022). Recent works (Magister et al., 2022; dhar et al., 2023; Ho et al., 2023) demonstrated that distillation on explanations generated from large models can increase the reasoning abilities of smaller models with ground truth filtering.

## 3 Method

The overview of our method is illustrated in Fig. 1: We are given a pre-trained Large Language Model (LLM) $M$ and a question-only train-

Table 1: Examples of 3 self-generated CoT reasoning paths given a question. Output 1 and 3 are the most consistent reasoning paths based on majority voting and kept as self-training data.

---

**Question**: Stefan goes to a restaurant with his family. They order an appetizer that costs $10 and 4 entrees that are $20 each. If they tip 20% of the total, what is the total amount of money that they spend?

---

**Multiple Path Decoding**:
**Output 1**: The appetizer costs $10. The entrees cost $20 each so in total 4 * $20 = $80. This means the total cost is $10 + $80 = $90. They tip 20% of it, so the total amount they spend is $90 * 1.2 = $108. The answer is 108.

**Output 2**: The appetizer costs $10 and the entrees are $20 each. There are 4 entrees so the sum is $20 * 4 = $80. The waiter gets 20% of the total. 20% of $80 is $80 * .2 = $16. The answer is $80 + $16 = $96. *(Incorrect reasoning path)*

**Output 3**: The appetizer costs $10. The entrees cost 4 * $20 = $80. The tip is 20% of the total, so it is 20% of the $90 they have spent. The tip is 0.2 * 90 = $18. The total they spent is $90 + $18 = $108. The answer is 108.

---

ing dataset $\mathcal{D}^{\texttt{train}} = \{x_i\}_{i=1}^{D}$ with few-shot Chain-of-Thought (CoT) examples (Wei et al., 2022c). We apply multiple path decoding with a sampling temperature $T > 0$ for generating $m$ reasoning paths and answers $\{r_{i_1}, r_{i_2}, \ldots, r_{i_m}\}$ for each question $x_i$ in $\mathcal{D}^{\texttt{train}}$, and use majority voting (self-consistency) to select the most consistent, highest confidence answer (Wang et al., 2022c). We then keep all reasoning paths that lead to the most consistent answer, apply mixed formats of prompts and answers for augmentation, and fine-tune the model on these self-generated reasoning-answer data. We consider our approach as making the model self-improve. In the following sections, we detail important designs within our method, along with additional approaches for the model to self-improve without supervised data.

### 3.1 Generating and Filtering Multiple Reasoning Paths

Self-consistency (Wang et al., 2022c) brings large improvements on reasoning tasks (e.g., $56.5\% \rightarrow 74.4\%$ on GSM8K test set), and the gap between greedy decoding and diverse decoding shows there is a potential for further improving the reasoning ability of $M$, using the self-selected high-confidence reasoning paths as training data.

For each training question $x_i$, we sample $m$ CoT reasoning paths, denoted as $\{r_{i_1}, r_{i_2}, \ldots, r_{i_m}\}$ (see Table 1 for examples). An example of a training question with the self-generated CoT reasoning paths is shown in Table 1. Since $M$ is

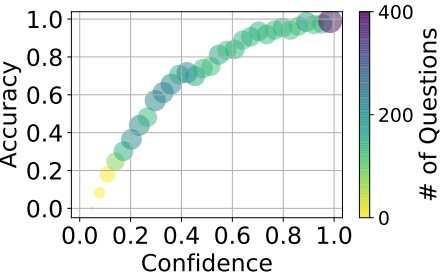

Figure 2: The relation of accuracy and confidence of the majority-voted answer after multiple path decoding on GSM8K training-set questions. A recent study (Kadavath et al., 2022) shows that language models are not perfectly-calibrated though their calibration increases with model size, and models with more than 10B parameters are reasonably calibrated on some few-shot tasks. This aligns well with our study and serve as the basis of this self-improving method.

prompted with the CoT examples from Wei et al. (2022c), we apply the same output parsing with "The answer is" to generate their predicted answers $\{y_{i_1}, y_{i_2}, \ldots, y_{i_m}\}$. The most consistent answer, which is not necessarily a correct answer, is selected by majority voting, denoted as $\tilde{y}_i = \arg\max_{y_{i_j}} \sum_{k=1}^{m} \mathbb{I}(y_{i_j} = y_{i_k})$. In Table 1, the most consistent answer $\tilde{y}$ is 108, derived by output path 1 and output path 3, while the output path 2 makes a mistake in calculating the cost of the foods. For all the training questions, we filter the CoT reasoning paths that reach $\tilde{y}$ as the final answer to be put into the self-training data,

Table 2: An example of how a reasoning path is augmented into four formats of training data with different prompts (in input) and answer styles (in output). Specifically, the *CoT prompting examples* used for each tasks are listed in Appendix A.2. The *Standard prompting examples* are the same question-answer pairs with *CoT prompting examples*, except that reasoning is removed.

---

**Question**: Amy is 10 years old. Jake is 8 years old. Alex's age is right in the middle. How old is Alex?
**Selected Chain-of-Thought**: Amy is 10 years old. Jake is 8 years old. Alex's age is in the middle of Amy and Jake, so Alex is ( 8 + 10 ) / 2 = 9 years old. The answer is 9.

---

**Mixed-formats of training data**:
**Format 1**: **Input**: *[CoT prompting examples]* + '\n' + *[Question]* + '\n' + 'A:'
**Output**: Amy is 10 years old. Jake is 8 years old. Alex's age is in the middle of Amy and Jake, so Alex is ( 8 + 10 ) / 2 = 9 years old. The answer is 9.

**Format 2**: **Input**: *[Standard prompting examples]* + '\n' + *[Question]* + '\n' + 'A:'
**Output**: The answer is 9.

**Format 3**: **Input**: *[Question]* + '\n' + 'A: Let's think step by step.'
**Output**: Amy is 10 years old. Jake is 8 years old. Alex's age is in the middle of Amy and Jake, so Alex is ( 8 + 10 ) / 2 = 9 years old. The answer is 9.

**Format 4**: **Input**: *[Question]* + '\n' + 'A:'
**Output**: The answer is 9.

---

denoted as $\mathcal{D}^{\texttt{self-consistent}} = \{x_i, \tilde{\boldsymbol{r}}_i\}$, where $\tilde{\boldsymbol{r}}_i = \{r_{i_j} | 1 \leq j \leq m, y_{i_j} = \tilde{y}_i\}$.

Since we do not use any ground truth labels to filter out cases where $\tilde{y}_i \neq y_i$, it is important that the self-generated CoT reasoning paths are mostly reliable and incorrect answers do not hurt the self-improvement of the model. We plot the relation between the accuracy and confidence of self-generated CoT paths for each question in GSM8K training set in Fig. 2. The confidence is the number of CoT paths leading to $\tilde{y}$ divided by the total path number $m$. The y-axis shows the accuracy of $\tilde{y}$ under a certain confidence. The circle area and the color darkness shows the number of questions under a certain confidence. We can observe that confident answers are more likely to be correct, which means that when a question has many consistent CoT paths, then the corresponding $\tilde{y}$ is more likely to be correct. On the other hand, when $\tilde{y}$ is wrong, it is likely to be supported by fewer CoT paths, and brings little noise to the training samples.

## 3.2 Training with Mixed Formats

To prevent the language model from overfitting to specific prompts or answer styles, we create four different formats for each reasoning path to be mixed in the self-training data, shown in Table 2. In the first format, a few Chain-of-Thought examples (questions followed by reasoning paths leading to the correct final answers) are prepended to the new question, while the language model output is trained to be the same with the filtered CoT reasoning paths. In the second format, we use examples of questions and their direct answers as standard prompting, and the language model output is supposed to also only contain the direct answer. The third and fourth format are similar to the first and second format, except that no example of question-answer pairs are given, so that the model will learn to think on its own in an in-context zero-shot manner. In the third format, where we want the model to output CoT reasoning without prepending examples containing CoT reasonings, we append "Let's think step by step." at the end of the input sequence, to guide the language model to generate step-by-step CoT reasoning paths (Kojima et al., 2022). The mixed formats of training samples are then used to fine-tune the pre-trained language model $M$.

## 3.3 Generating Questions and Prompts

In some cases where even training questions or human-curated CoT prompts are limited, our method may not generate sufficient training samples for language model self-training. Therefore,

we investigate how to self-generate more training questions as well as example prompts to further reduce human effort.

**Question Generation.** Previous work (Yoo et al., 2021; Meng et al., 2022) discuss few-shot data augmentation by generating diverse training samples using LLMs. However, those methods are designed for classification tasks and require ground truth label for each few-shot example. We use a simple yet effective approach to generate diverse questions (without using ground truth answers) from a few example questions. Specifically, we randomly sample and concatenate example questions in a random order as input prompt, and let the language model generate consecutive sequences as new questions. We repeat the process to obtain a large set of new questions, then use self-consistency (Wang et al., 2022c) to only keep the questions that have a highly confident answer. Those questions are then used as self-generated training questions.

**Prompt Generation.** Given a set of questions, humans can write CoT examples as reasoning paths leading to the final answer. In zero-shot setting without manual prompts, we can generate these CoT paths using the model itself. Following (Kojima et al., 2022), we start the answer with "A: Let's think step by step." and let the language model generate the consecutive reasoning paths. We then use those generated reasoning paths as examples for few-shot CoT prompting.

## 4 Experimental Setup

**Tasks and Datasets.** We demonstrate the effectiveness of our method on three types of tasks[1]:

- **Arithmetic reasoning**: We use the math problem set GSM8K (Cobbe et al., 2021), and a reading comprehension benchmark DROP (Dua et al., 2019) which requires numerical reasoning. We follow (Zhou et al., 2022) to partition the DROP dataset into football related and non-football related subsets for training.

- **Commonsense reasoning**: We use the Open-BookQA (Mihaylov et al., 2018) dataset, and the AI2 Reasoning Challenge (ARC) (Clark et al., 2018) dataset. Note that for ARC, we only use

the Challenge sub-set (ARC-c) in our experiments. Both datasets contain multiple-choice questions.

- **Natural Language Inference**: We use the Adversarial NLI (ANLI) (Mihaylov et al., 2018) subsets, ANLI-A2 and ANLI-A3, which are the more challenging subsets compared to ANLI-A1. These datasets contain pairs of sentences with relations of entailment, neutral, or contradiction.

**Models, Training settings and Hyperparameters.** We follow previous studies (Wei et al., 2022c; Wang et al., 2022c) and conduct our experiments on the PaLM 540B model (Chowdhery et al., 2022), an autoregressive Transformer-based language model. The CoT examples for each dataset are listed in Appendix A.2. We generate $m = 32$ reasoning paths for each question in a training set, followed by format augmentation in Sec. 3.2. For DROP and ANLI-A2/A3, we sample 5k examples for reasoning path generation to reduce the training burden; For other datasets, we keep the whole training set. For each dataset, we fine-tune the model for 10k steps with a learning rate of $5e-5$ and a batch size of 32. We use a sampling temperature of $T = 0.7$ with the pre-trained model as suggested by (Wang et al., 2022c). We use $T = 1.2$ for the language model after self-improvement (**LMSI**). We set the maximum number of decoded steps to 256 for all experiments.

## 5 Experiments and Results

We conduct a series of experiments to demonstrate the effectiveness of our proposed self-improving method. First, we apply our method on each individual dataset (task) and report the results. We then merge the generated data from all datasets and train one model to study the generalization ability of the model on unseen datasets as in (Wei et al., 2021). In addition to the results of using generated CoT reasoning paths, we show studies on generating input questions and few-shot prompts. We end with ablation studies on model sizes and hyperparameters.

### 5.1 Main Results

We list the results of using the 540B PaLM model before and after **LMSI** in Table 3. For each model, during test time, we apply three separate prompting methods on all six datasets: standard-prompting, CoT-Prompting, and Self-Consistency. We observe

---

[1]We evaluate on the test sets of GSM8K, ARC, Open-BookQA, and ANLI, and the dev set of DROP (ground truth labels of the test set are not publicly available).

Table 3: Accuracy results on six reasoning benchmarks with or without **LMSI** using different prompting method.

| Prompting Method | w. or w/o LMSI | GSM8K | DROP | ARC-c | OpenBookQA | ANLI-A2 | ANLI-A3 |
|---|---|---|---|---|---|---|---|
| Standard-Prompting | w/o **LMSI** | 17.9 | 60.0 | 87.1 | 84.4 | 55.8 | 55.8 |
| | w. **LMSI** | 32.2 (+14.3) | 71.7 (+11.7) | 87.2 (+0.1) | 92.0 (+7.6) | 64.8 (+9.0) | 66.9 (+11.1) |
| CoT-Prompting | w/o **LMSI** | 56.5 | 70.6 | 85.2 | 86.4 | 58.9 | 60.6 |
| | w. **LMSI** | 73.5 (+17.0) | 76.2 (+5.6) | 88.3 (+3.1) | 93.0 (+6.6) | 65.3 (+6.4) | 67.3 (+6.7) |
| Self-Consistency | w/o **LMSI** | 74.4 | 78.2 | 88.7 | 90.0 | 64.5 | 63.4 |
| | w. **LMSI** | **82.1** (+7.7) | **83.0** (+4.8) | **89.8** (+1.1) | **94.4** (+4.4) | **66.5** (+2.0) | **67.9** (+4.5) |

Table 4: Comparison of CoT-prompting accuracy results on six Out-Of-Domain benchmarks with or without training on six In-Domain (GSM8K, DROP, ARC-c, OpenBookQA, ANLI-A2, ANLI-A3) training-set questions.

| | Self-training data | AQUA | SVAMP | StrategyQA | ANLI-A1 | RTE | MNLI-M/MM |
|---|---|---|---|---|---|---|---|
| w/o **LMSI** | - | 35.8 | 79.0 | 75.3 | 68.8 | 79.1 | 72.0/74.0 |
| w. **LMSI** | GSM8K + DROP + ... | 39.0 (+3.2) | 82.8 (+3.8) | 77.8 (+2.5) | 79.2 (+10.4) | 80.1 (+1.0) | 81.8/82.2 (+9.8/+8.2) |

that after **LMSI** , the performance of all three prompting methods increase by a large margin. We observe significant improvement, comparing self-consistency versus **LMSI** with self-consistency: +7.7% on GSM8K, +4.8% on DROP, +4.4% on OpenBookQA, and +4.5% on ANLI-A3. This shows that our proposed method is quite effective. Furthermore, the single path CoT-Prompting performance of **LMSI** is close to or even better than the multiple path Self-Consistency performance of the model without **LMSI** , showing that **LMSI** truly helps the language model learn from the multiple consistent reasoning paths. We also apply **LMSI** on a recently proposed public language model, UL2 (20B) (Tay et al., 2022), and show the results in Appendix A.1. Compared to the 540B PaLM model (decoder-only), UL2 has a smaller scale, and a different architecture (encoder-decoder). We observe that for most datasets, **LMSI** still outperforms the original UL2 results, but the improvement is not as large as that on the 540B PaLM model.

**Multi-task self-training for unseen tasks.** To demonstrate the generalization ability of **LMSI** , we conduct experiments of self-training on a mixture of the training-set questions from the above six datasets (denoted as In-Domain tasks), then use the same model checkpoint for the evaluation on six Out-Of-Domain (OOD) tasks, as shown in Table 4. Of all the OOD tasks: (1) **AQUA** (Ling et al., 2017b) and **SVAMP** (Patel et al., 2021) are arithmetic reasoning tasks; (2) **StrategyQA** (Geva et al., 2021) is a commonsense reasoning task; (3) **ANLI-A1** (Nie et al., 2019), **RTE** (Dagan et al., 2005) and **MNLI-M/MM** (Williams et al., 2018) are nat-

ural language inference tasks.[2] Among these tasks, **AQUA**, **StrategyQA**, and **RTE** are significantly different from any In-Domain task, and have their own few-shot prompts. From Table 4, we observe that **LMSI** achieves higher accuracy results on all OOD tasks, showing that the overall reasoning ability of the language model is improved.

**Importance of training with augmented formats.** We demonstrate the importance of training language models with augmented formats (both Chain-of-Thought prompting and direct prompting, and both few-shot prompting and zero-shot prompting). In Table 5, we list the results of **LMSI** with all four formats, the results of **LMSI** with only direct answer formats, and the results of **LMSI** with only few-shot Chain-of-Thought prompting formats. The results show that without the CoT formats, the language model can still self-improve, but the performance gain drops by a large amount compared to using all four formats. However, if only using few-shot CoT prompting format for self-training, the model can overfit to the prompting style and may not generalize well on downstream tasks.

### 5.2 Pushing the limit of self-improvements

**Self-Generating Questions** We further explore the few-shot setting where there are only limited training questions in the target domain. On GSM8K, we sample 10 real questions as few-shot

---

[2]We evaluate on the test set of SVAMP and ANLI, the dev set of MNLI and RTE (ground truth labels of the test sets are not publicly available). For StrategyQA we use the question-only set from (bench collaboration, 2022).

Table 5: Ablation study: **LMSI** with different combinations of training format on GSM8K dataset.

| | Results on **GSM8K** | |
| --- | --- | --- |
| | Std. Prompting | CoT Prompting |
| w/o **LMSI** | 17.9 | 56.5 |
| **LMSI** w/o CoT formats | 23.6 (+5.7) | 61.6 (+5.1) |
| **LMSI** only few-shot CoT | 29.2 (+11.3) | 69.4 (+12.9) |
| **LMSI** w/ CoT formats | 32.2 (+14.3) | 73.5 (+17.0) |

samples, and use the language model to generate more training questions using the method in Section 3.3. We then self-train the language model with these generated questions and list the results in Table 6. The results show that using self-generated questions still improves the reasoning ability of language models, but using the real training-set questions leads to better results.

Table 6: Accuracy on GSM8K test set after self-training on different question sets. Results are shown for both CoT-Prompting (CoT) and Self-Consistency (SC).

| Questions used for Self-Training | **GSM8K** | |
| --- | --- | --- |
| | CoT | SC |
| w/o **LMSI** | - | 56.5 | 74.4 |
| w. **LMSI** Generated | 66.2 (+9.7) | 78.1 (+3.7) |
| w. **LMSI** Training-set | 73.5 (+17.0) | 82.1 (+7.7) |

**Self-Generating Few-Shot CoT Prompts.** We explore the situation where no in-domain CoT examples are provided for a task. We apply the Step-by-Step method (Kojima et al., 2022) to generate CoT examples using the language model as described in Section 3.3, and show the results in Figure 3. We observe that few-shot prompting with self-generated Step-by-Step CoT examples substantially outperforms the Step-by-Step (Kojima et al., 2022) baseline (66.2% vs 53.8% at 10 paths, 74.2% vs 70.1% at 40 paths), and nearly matches the performance of human-written few-shot CoT (Wei et al., 2021) (74.4% at 40 paths (Wang et al., 2022c)). The strong performance of "Few-Shot w/ Step-by-Step" despite the limited accuracy of prompt examples (43.0% for greedy Step-by-Step) likely comes from leveraging more diverse CoT prompts for multi-path decoding (Li et al., 2022b), where at 40 paths it uses 20 generate prompt-templates, each with 4-shot CoT examples, i.e. a total of 80 generated CoT examples compared to 8 human-written examples use in Wei et al. (2022c).

Since we did not use training questions or few-shot CoT examples, 74.2% also marks the new state-of-the-art zero-shot performance on GSM8K.

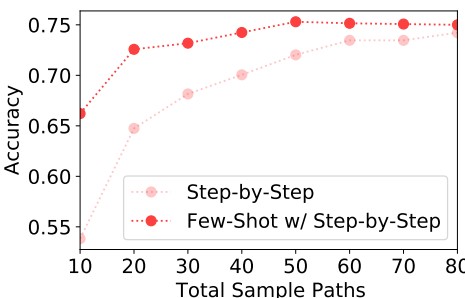

Figure 3: Accuracy results on GSM8K test set using 540B model with multi-path sampling and self-consistency (Wang et al., 2022c). "Step-by-Step" is the baseline performance of Kojima et al. (2022) plus self-consistency (Wang et al., 2022c), while our "Few-Shot w/ Step-by-Step" uses exemplars self-generated from Step-by-Step (greedy decoding) for few-shot prompting the LLM.

### 5.3 Distillation to smaller models

Table 7: Distillation from 540B model to small models. We see that distilled smaller models outperform models that are one-tier larger.

| | Results on **GSM8K** | | |
| --- | --- | --- | --- |
| | 8 billion | 62 billion | 540 billion |
| w/o **LMSI** | 5.0 | 29.7 | 56.5 |
| Distilled from **LMSI** | 33.4 (+28.4) | 57.4 (+27.7) | - |

We also explore whether the knowledge can be distilled to smaller models, such as in distillation (Hinton et al., 2015) and in Zelikman et al. (2022). We use the same set of training samples generated by the 540B PaLM model, but fine-tune on models with smaller sizes (8B PaLM model and 62B PaLM model respectively), and show the results of CoT-prompting in Table 7. It is interesting to point out that after distillation from **LMSI**, the 62B model can outperform the pre-trained 540B model, and the 8B model can outperform the pre-trained 62B model. This implies that for downstream applications with limited computing resources, the reasoning knowledge from large models can be used to largely enhance small models to achieve competitive performance.

### 5.4 Hyperparameter Studies

**Sampling Temperature after Self-Improvement.** We study the effect of varying the temperature $T$

for multiple path decoding after **LMSI** is applied. Specifically, we vary $T$ between $[0.7, 1.0, 1.2, 1.5]$ and show the results on GSM8K and DROP dataset respectively in Fig. 4. As shown in the figure, $T = 1.2$ benefits both datasets the most, and is used in the Self-Consistency method for **LMSI** on all datasets. We notice that the optimal $T$ after model self-improvement is larger than the optimal $T = 0.7$ (Wang et al., 2022c) before self-improvement. We believe the reason is that after training the model, the entropy of the output distribution is reduced.

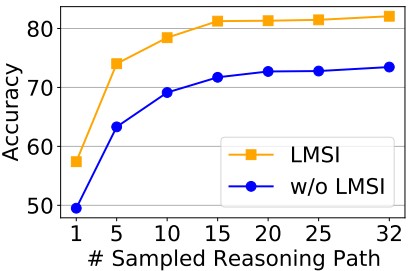

Figure 5: Accuracy results with or without **LMSI** on GSM8K test set using different numbers of sampled reasoning path for Self-Consistency.

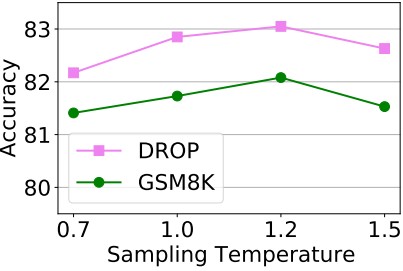

Figure 4: Accuracy results of **LMSI** on GSM8K and DROP test set when different sampling temperatures are applied for Self-Consistency.

**Number of Sampled Reasoning Paths.** We study whether the number of sampled reasoning paths $m$ for Self-Consistency largely affects the accuracy after **LMSI** is applied. We show the accuracy on GSM8K test set for models both with or without **LMSI** in Fig. 5. For both cases, setting $m = 15$ already achieves a reasonably good accuracy, and using a larger $m$ only brings marginal improvements. We also notice that after Self-Improvement, using 5 paths for Self-Consistency can already surpass the performance of using 32 paths for model without Self-Improvement. Thus, with a well-improved model, huge computing resources can be saved when applied to real applications.

## 6 Conclusions

We demonstrated that a Large Language Model (LLM) is capable of improving its performance on reasoning datasets by training on its own generated labels, given input questions only. Experiments using the PaLM model with 540 billion parameters show that **LMSI** improves the accuracy scores by 1.1% to 7.7% on six datasets, without training on ground truth labels. Furthermore, we show that

it is possible for the LLM to self-improve even on its own generated questions and few-shot CoT prompts. As part of our future work, we plan to combine large-scale generated data from **LMSI** and existing supervised data, to further improve the performance of LLMs.

## Limitations

Our approach mainly relies on the effectiveness of demonstration-based in-context few-shot learning which works most effectively on large language models, according to Wei et al. (2022a). For example, Zelikman et al. (2022) showed that a 6B model, GPT-J, achieves only 3.1% accuracy on GSM8K with few-shot CoT prompting, while GPT-3 (175 B) achieves 46.9%, according to Wei et al. (2022c). Moreover, a recent study (Kadavath et al., 2022) shows that language model calibration increases with model size. This aligns well with our observations that larger models are better at self-improving. Based on these existing studies, we believe that **LMSI** is more applicable to large-scale language models. In addition, we show that distillation from large models to small models are very promising in Sec. 5.3. Therefore, smaller models can also be improved when large model APIs are accessible. We are fortunate to have enough resources for this work. Though the computation requirements for training large-scale language models are still prohibitively high for most researchers to conduct empirical studies along this line, we believe that our findings are conceptually useful for the NLP community by providing new insights for the properties of large language models.

## Acknowledgments

We thank anonymous reviewers for valuable and insightful feedback.

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

# A Appendix

## A.1 Results on UL2 model

We also apply **LMSI** on a recently proposed public language model, UL2 (Tay et al., 2022), using the pre-trained model at step 2,650,000[3]. We use a fixed set of hyperparameters for fine-tuning on each dataset. Specifically, we generate $m = 40$ reasoning paths for each question in a training set for majority voting. We fine-tune the model for 10k steps with a learning rate of $5e-5$ and a batch size of 32. For multiple path decoding, we use a sampling temperature of $T = 0.5$ with the pre-trained UL2 model following Tay et al. (2022), and set $T = 0.7$ for the language model after **LMSI** . We set the maximum number of decode steps to 256 for all experiments.

The results are shown in Table 8. For arithmetic reasoning datasets, we follow (Tay et al., 2022) to provide both exact matching accuracy scores as well as accuracy scores after an equation-correction postprocessing step. We observe that for most datasets, **LMSI** still improves the reasoning accuracy (+1.6% on DROP, +1.2% on OpenBookQA, and +0.7% on ANLI-A2), but the improvement on UL2 is not as large as that on 540B. We think the reason is that, since **LMSI** exploits the implicit rationale of language models, and the capacity of a language model is determined by its size, larger models can capture more high-order semantics and are more likely to benefit from **LMSI** . For example, on the adversarial entailment tasks of ANLI (which is a three-class classification problem with labels "yes", "no", or "it is not possible to tell"), the UL2 model w/o **LMSI** only achieves an accuracy of marginally above $1/3$, implying that the model is slightly better than doing random guess on this challenging task without any training. Our proposed **LMSI** can still improve the performance under this hard case by training on its implicit knowledge from self-generated paths.

Table 8: Accuracy results on six reasoning benchmarks with **LMSI** on UL2. On GSM8K and DROP, we also include accuracy scores after an equation-correction postprocessing step.

|  | **Prompting Method** | **GSM8K** | **DROP** | **ARC**-c | **OpenBookQA** | **ANLI**-A2 | **ANLI**-A3 |
|---|---|---|---|---|---|---|---|
| w/o **LMSI** | CoT-Prompting | 5.4/7.1 | 11.1/16.8 | 49.9 | 53.6 | 35.9 | 33.8 |
|  | Self-Consistency | 6.4/9.9 | 16.8/26.5 | 54.7 | 54.0 | 37.4 | 36.8 |
| **LMSI** | CoT-Prompting | 6.1/8.6 | 11.4/17.1 | 50.9 | 53.8 | 35.4 | 34.4 |
|  | Self-Consistency | 7.9/10.2 | 18.1/28.1 | 54.9 | 55.2 | 38.1 | 37.4 |

## A.2 Chain-of-Thought Prompts for Each Dataset

We list the Chain-of-Thought Prompts for each dataset for "CoT-Prompting" experiments and self-generated training samples.

---

[3]UL2: https://github.com/google-research/google-research/tree/master/ul2

Table 9: Few-shot CoT prompts for GSM8K and SVAMP, from (Wei et al., 2022c).

---

**Q:** There are 15 trees in the grove. Grove workers will plant trees in the grove today. After they are done, there will be 21 trees. How many trees did the grove workers plant today?

**A:** We start with 15 trees. Later we have 21 trees. The difference must be the number of trees they planted. So, they must have planted 21 - 15 = 6 trees. The answer is 6.

**Q:** If there are 3 cars in the parking lot and 2 more cars arrive, how many cars are in the parking lot?

**A:** There are 3 cars in the parking lot already. 2 more arrive. Now there are 3 + 2 = 5 cars. The answer is 5.

**Q:** Leah had 32 chocolates and her sister had 42. If they ate 35, how many pieces do they have left in total?

**A:** Leah had 32 chocolates and Leah's sister had 42. That means there were originally 32 + 42 = 74 chocolates. 35 have been eaten. So in total they still have 74 - 35 = 39 chocolates. The answer is 39.

**Q:** Jason had 20 lollipops. He gave Denny some lollipops. Now Jason has 12 lollipops. How many lollipops did Jason give to Denny?

**A:** Jason had 20 lollipops. Since he only has 12 now, he must have given the rest to Denny. The number of lollipops he has given to Denny must have been 20 - 12 = 8 lollipops. The answer is 8.

**Q:** Shawn has five toys. For Christmas, he got two toys each from his mom and dad. How many toys does he have now?

**A:** He has 5 toys. He got 2 from mom, so after that he has 5 + 2 = 7 toys. Then he got 2 more from dad, so in total he has 7 + 2 = 9 toys. The answer is 9.

**Q:** There were nine computers in the server room. Five more computers were installed each day, from monday to thursday. How many computers are now in the server room?

**A:** There are 4 days from monday to thursday. 5 computers were added each day. That means in total 4 * 5 = 20 computers were added. There were 9 computers in the beginning, so now there are 9 + 20 = 29 computers. The answer is 29.

**Q:** Michael had 58 golf balls. On tuesday, he lost 23 golf balls. On wednesday, he lost 2 more. How many golf balls did he have at the end of wednesday?

**A:** Michael initially had 58 balls. He lost 23 on Tuesday, so after that he has 58 - 23 = 35 balls. On Wednesday he lost 2 more so now he has 35 - 2 = 33 balls. The answer is 33.

**Q:** Olivia has $23. She bought five bagels for $3 each. How much money does she have left?

**A:** She bought 5 bagels for $3 each. This means she spent 5 * $3 = $15 on the bagels. She had $23 in beginning, so now she has $23 - $15 = $8. The answer is 8.

---

Table 10: Few-shot CoT prompts for OpenBookQA, from (Wang et al., 2022b).

---

**Q:** Poison causes harm to which of the following? (a) a Tree (b) a robot (c) a house (d) a car

**A:** Poison will harm living things, only a tree is a living thing. The answer is (a).

**Q:** As you look deeper into a Marbel you can see (a) the future (b) minut defects (c) colors (d) the other side

**A:** Marbel is not transparent, so you can not see the other side. Marbel does not necessarily have multiple colors. You will see minut defects. The answer is (b).

**Q:** When food is reduced in the stomach (a) the mind needs time to digest (b) take a second to digest what I said (c) nutrients are being deconstructed (d) reader's digest is a body of works

**A:** The food is being deconstructed in the stomach during digestion. The answer is (c).

**Q:** The sun is responsible for (a) puppies learning new tricks (b) children growing up and getting old (c) flowers wilting in a vase (d) plants sprouting, blooming and wilting

**A:** The sun can affect the growing of living things, like plants. The answer is (d).

---

Table 11: Few-shot CoT prompts for DROP (nonfootball), from (Zhou et al., 2022).

**Q:** Since the 1970s, U.S. governments have negotiated managed-trade agreements, such as the North American Free Trade Agreement in the 1990s, the Dominican Republic-Central America Free Trade Agreement in 2006, and a number of bilateral agreements. In Europe, six countries formed the European Coal and Steel Community in 1951 which became the European Economic Community in 1958. Two core objectives of the EEC were the development of a common market, subsequently renamed the single market, and establishing a customs union between its member states. How many years did the European Coal and Steel Community exist?

**A:** According to the passage, the European Coal and Steel Community was established in 1951 and became the EEC in 1958. 1958 - 1951 = 7. So the answer is 7.

**Q:** In the county, the population was spread out with 23.50% under the age of 18, 8.70% from 18 to 24, 29.70% from 25 to 44, 24.70% from 45 to 64, and 13.30% who were 65 years of age or older. How many more percent are under the age of 18 compared to the 18 to 24 group?

**A:** According to the passage, 23.5% are under the age of 18, and 8.7% are from ages 18 to 24. 23.5% - 8.7% = 14.8%. So the answer is 14.8.

**Q:** Playing in their second straight Thanksgiving game, the Eagles struggled especially on defense, where they were unable to stop the much-hyped Lions offense. The worst of it all was how unproven rookie Eric Rowe was tasked with covering wide receiver Calvin Johnson, leading to Johnson catching 3 touchdowns. Stafford's five passing touchdowns, including three of them to Johnson was too much for the Eagles to overcome and for the second consecutive time this season, the Eagles gave up 45 points in a game. With the loss, the Eagles drop to 4-7 on the season and 6-1 when playing on Thanksgiving. How many TD passes did Stafford throw other than to Johnson?

**A:** According to the passage, Stafford threw 5 TD passes, 3 of which were to Johnson. 5 - 3 = 2. So the answer is 2.

Table 12: Few-shot CoT prompts for DROP (football), from (Zhou et al., 2022).

**Q:** The Seahawks played the San Francisco 49ers. In the first quarter, the Hawks RB Julius Jones got a 27-yard TD run, along with DT Craig Terrill returning a fumble 9 yards for a touchdown. In the third quarter, the 49ers almost rallied as RB H. J. Torres made a 12-yard TD pass to Lucas Nelly, along with Mare kicking a 32-yard field goal. In the final quarter, Julius Jones got another 11-yard TD. How many yards do the shortest touchdown run and the longest touchdown pass combine for?

**A:** All the touchdown runs are: a 27-yard touchdown run, a 9-yard touchdown run, a 11-yard touchdown run. The smallest number among 27, 9, 11 is 9. So the shortest touchdown run was 9 yards. All the touchdown passes are: a 12-yard touchdown pass. So the longest touchdown pass was 12 yards. So the shortest touchdown run and the longest touchdown pass combine for 9 + 12 = 21 yards. So the answer is 21 yards.

**Q:** The Steelers went home for a duel with the Baltimore Ravens. Pittsburgh would deliver the opening punch in the first quarter with a 1-yard touchdown from running back Rashard Mendenhall. The Ravens would make it even as running back Willis McGahee got a 9-yard TD. The Ravens kicker Billy Cundiff got a 45-yard field goal in the second quarter, concluding the first half with a 10-7 lead. The Steelers brought the game into overtime with a 38-yard field goal by Andrew Foster. The Ravens Billy Cundiff pulled off a winning 33-yard field goal in overtime. How many points did the Ravens have at halftime?

**A:** The Ravens kicker Billy Cundiff got a 45-yard field goal in the second quarter, concluding the first half with a 10-7 lead. So the Ravens had 10 points at halftime. So the answer is 10 points.

**Q:** The Vikings flew to Bank of America Stadium to face the Carolina Panthers. After a scoreless first quarter, Carolina got on the board with quarterback Matt Moore finding fullback Brad Hoover on a 1-yard TD pass. After yet another scoreless quarter, Carolina sealed the game as Matt Moore completed a 42-yard touchdown pass to wide receiver Steve Smith. How many scoreless quarters were there?

**A:** The first and third quarters were the scoreless quarters. So there are 2 scoreless quarters. So the answer is 2.

Table 13: Few-shot CoT prompts for NLI tasks, including ANLI and MNLI, from (Wang et al., 2022b).

Premise:
"Conceptually cream skimming has two basic dimensions - product and geography."
Based on this premise, can we conclude the hypothesis "Product and geography are what make cream skimming work." is true?
OPTIONS:
- yes
- no
- it is not possible to tell

A: Based on "cream skimming has two basic dimensions" we can't infer that these two dimensions are what make cream skimming work. The answer is it is not possible to tell.

Premise:
"One of our member will carry out your instructions minutely."
Based on this premise, can we conclude the hypothesis "A member of my team will execute your orders with immense precision." is true?
OPTIONS:
- yes
- no
- it is not possible to tell

A: "one of" means the same as "a member of", "carry out" means the same as "execute", and "minutely" means the same as "immense precision". The answer is yes.

Premise:
"Fun for adults and children."
Based on this premise, can we conclude the hypothesis "Fun for only children." is true?
OPTIONS:
- yes
- no
- it is not possible to tell

A: "adults and children" contradicts "only children". The answer is no.

Premise:
"He turned and smiled at Vrenna."
Based on this premise, can we conclude the hypothesis "He smiled at Vrenna who was walking slowly behind him with her mother." is true?
OPTIONS:
- yes
- no
- it is not possible to tell

A: the premise does not say anything about "Vrenna was walking". The answer is it is not possible to tell.

Premise:
"well you see that on television also"
Based on this premise, can we conclude the hypothesis "You can see that on television, as well." is true?
OPTIONS:
- yes
- no
- it is not possible to tell

A: "also" and "as well" mean the same thing. The answer is yes.

Premise:
"Vrenna and I both fought him and he nearly took us."
Based on this premise, can we conclude the hypothesis "Neither Vrenna nor myself have ever fought him." is true?
OPTIONS:
- yes
- no
- it is not possible to tell

A: "Vrenna and I both" contradicts "neither Vrenna nor myself". The answer is no.

Table 14: Few-shot CoT prompts for ARC-challenge, from (Wang et al., 2022c).

---

**Q:** George wants to warm his hands quickly by rubbing them. Which skin surface will produce the most heat? (a) dry palms. (b) wet palms. (c) palms covered with oil. (d) palms covered with lotion.

**A:** Dry surfaces will more likely cause more friction via rubbing than other smoother surfaces, hence dry palms will produce the most heat. The answer is (a).

**Q:** Which factor will most likely cause a person to develop a fever? (a) a leg muscle relaxing after exercise. (b) a bacterial population in the bloodstream. (c) several viral particles on the skin. (d) carbohydrates being digested in the stomach.

**A:** Option (b), bacterial population is the most likely cause for a person developing fever. The answer is (b).

**Q:** Which change in the state of water particles causes the particles to become arranged in a fixed position? (a) boiling. (b) melting. (c) freezing. (d) evaporating.

**A:** When water is freezed, the particles are arranged in a fixed position; the particles are still moving for all other options. The answer is (c).

**Q:** When a switch is used in an electrical circuit, the switch can (a) cause the charge to build. (b) increase and decrease the voltage. (c) cause the current to change direction. (d) stop and start the flow of current.

**A:** The function of a switch is to start and stop the flow of a current. The answer is (d).

---

Table 15: Few-shot CoT prompts for AQUA, from (Wang et al., 2022c).

---

**Q:** John found that the average of 15 numbers is 40. If 10 is added to each number then the mean of the numbers is? Answer Choices: (a) 50 (b) 45 (c) 65 (d) 78 (e) 64

**A:** If 10 is added to each number, then the mean of the numbers also increases by 10. So the new mean would be 50. The answer is (a).

**Q:** If a / b = 3/4 and 8a + 5b = 22,then find the value of a. Answer Choices: (a) 1/2 (b) 3/2 (c) 5/2 (d) 4/2 (e) 7/2

**A:** If a / b = 3/4, then b = 4a / 3. So 8a + 5(4a / 3) = 22. This simplifies to 8a + 20a / 3 = 22, which means 44a / 3 = 22. So a is equal to 3/2. The answer is (b).

**Q:** A person is traveling at 20 km/hr and reached his destiny in 2.5 hr then find the distance? Answer Choices: (a) 53 km (b) 55 km (c) 52 km (d) 60 km (e) 50 km

**A:** The distance that the person traveled would have been 20 km/hr * 2.5 hrs = 50 km. The answer is (e).

**Q:** How many keystrokes are needed to type the numbers from 1 to 500? Answer Choices: (a) 1156 (b) 1392 (c) 1480 (d) 1562 (e) 1788

**A:** There are 9 one-digit numbers from 1 to 9. There are 90 two-digit numbers from 10 to 99. There are 401 three-digit numbers from 100 to 500. 9 + 90(2) + 401(3) = 1392. The answer is (b).

---

Table 16: Few-shot CoT prompts for StrategyQA, from (Wei et al., 2022c).

**Q:** Do hamsters provide food for any animals?

**A:** Hamsters are prey animals. Prey are food for predators. Thus, hamsters provide food for some animals. The answer is yes.

**Q:** Could Brooke Shields succeed at University of Pennsylvania?

**A:** Brooke Shields went to Princeton University. Princeton University is about as academically rigorous as the University of Pennsylvania. Thus, Brooke Shields could also succeed at the University of Pennsylvania. The answer is yes.

**Q:** Yes or no: Hydrogen's atomic number squared exceeds number of Spice Girls?

**A:** Hydrogen has an atomic number of 1. 1 squared is 1. There are 5 Spice Girls. Thus, Hydrogen's atomic number squared is less than 5. The answer is no.

**Q:** Yes or no: Is it common to see frost during some college commencements?

**A:** College commencement ceremonies can happen in December, May, and June. December is in the winter, so there can be frost. Thus, there could be frost at some commencements. The answer is yes.

**Q:** Yes or no: Could a llama birth twice during War in Vietnam (1945-46)?

**A:** The War in Vietnam was 6 months. The gestation period for a llama is 11 months, which is more than 6 months. Thus, a llama could not give birth twice during the War in Vietnam. The answer is no.

**Q:** Yes or no: Would a pear sink in water?

**A:** The density of a pear is about 0.6 $g/cm^3$, which is less than water. Objects less dense than water float. Thus, a pear would float. The answer is no.

---

Table 17: Few-shot CoT prompts for RTE, from (Wang et al., 2022b).

Premise:
"No Weapons of Mass Destruction Found in Iraq Yet."
Based on this premise, can we conclude the hypothesis "Weapons of Mass Destruction Found in Iraq." is true?

**A:** "No Weapons of Mass Destruction Found" contradicts "Weapons of Mass Destruction Found". The answer is no.

Premise:
"A place of sorrow, after Pope John Paul II died, became a place of celebration, as Roman Catholic faithful gathered in downtown Chicago to mark the installation of new Pope Benedict XVI."
Based on this premise, can we conclude the hypothesis "Pope Benedict XVI is the new leader of the Roman Catholic Church." is true?'

**A:** "installation of new Pope Benedict XVI." means "Pope Benedict XVI is the new leader". The answer is yes.

Premise:
"A man is due in court later charged with the murder 26 years ago of a teenager whose case was the first to be featured on BBC One's Crimewatch. Colette Aram, 16, was walking to her boyfriend's house in Keyworth, Nottinghamshire, on 30 October 1983 when she disappeared. Her body was later found in a field close to her home. Paul Stewart Hutchinson, 50, has been charged with murder and is due before Nottingham magistrates later."
Based on this premise, can we conclude the hypothesis "Paul Stewart Hutchinson is accused of having stabbed a girl." is true?

**A:** The premise does not say Paul Stewart Hutchinson "stabbed" this girl. The answer is no.

Premise:
"Herceptin was already approved to treat the sickest breast cancer patients, and the company said, Monday, it will discuss with federal regulators the possibility of prescribing the drug for more breast cancer patients."
Based on this premise, can we conclude the hypothesis "Herceptin can be used to treat breast cancer." is true?

**A:** "Herceptin was approved to treat breast cancer" implies that "Herceptin can be used to treat breast cancer". The answer is yes.