# OpenReview forum: "Large Language Models Can Self-Improve"
_EMNLP/2023/Conference — EMNLP 2023 Main_

### Official Review · Reviewer_g6YS · 2023-08-04

**Typos Grammar Style And Presentation Improvements:** N.A.
**Soundness:** 3

**Excitement:**

3: Ambivalent: It has merits (e.g., it reports state-of-the-art results, the idea is nice), but there are key weaknesses (e.g., it describes incremental work), and it can significantly benefit from another round of revision. However, I won't object to accepting it if my co-reviewers champion it.

**Missing References:**

N.A.

**Paper Topic And Main Contributions:**

This paper explores the potential of improving large language models (LLMs) through training with only unlabelled data. The authors utilized a specific LLM called PaLM and generated multiple possible answers for a given problem. They selected answers where the model exhibited high confidence and used them as target outputs to retrain the LLM. The results showed that this approach outperformed the non-trained version, indicating that unlabelled fine-tuning can lead to self-improvement of LLMs. The proposed method involves generating candidates (high-confidence rationale-augmented answers) for unlabelled questions and then fine-tuning the LLM on these self-generated solutions. This process significantly enhanced the performance of a large 540B model on various datasets, including GSM8K, DROP, OpenBookQA, and ANLI, which are commonly used for evaluation in the research community.

Main contributions:
1. Large language models can enhance their performance without ground truth outputs by utilizing CoT reasoning and self-consistency, resulting in significant improvements in in-domain multi-task performances and out-of-domain generalization.
2. The study includes detailed ablation experiments on training sample formatting and sampling temperature after fine-tuning, leading to the identification of critical design choices for successful self-improvement by LLMs.
3. The authors propose two novel approaches for model self-improvement under extreme low-resource scenarios, achieving an impressive 74.2% accuracy on zero-shot GSM8K, outperforming previous methods by Kojima et al. (2022) and their naive extension with Wang et al. (2022b).

**Questions For The Authors:**

1. Have you tested the suggested techniques on other large language models, such as LLaMA? Can we expect these methods to retain their effectiveness when applied to other models?

**Reasons To Accept:**

1. The empirical results demonstrate the effectiveness of the proposed method.
2. The approach is relatively straightforward, although it becomes challenging to estimate the impact of various hyperparameters and choices (some of which are explored later in the paper) at each stage due to the complexity of fine-tuning a large model like this.
3. The process of distillation, specifically fine-tuning a smaller model based on the generations of the 540B model, proves to be effective.

**Reasons To Reject:**

1. Prompting has become a widely recognized method for interacting with Large Language Models (LLMs) due to the considerable size and inefficiency or impossibility of training them in many cases. Historically, various strategies have been devised to enhance the performance of LLMs substantially, including methodologies like Chain of Thoughts and step-by-step processes. One intuitive extension has been the generation of multiple outcomes using identical or varied strategies, followed by selecting the more confident response or clustering them or employing majority voting. This paper amalgamates previous methods and introduces a straightforward extension that involves training on confident samples. This concept, although unoriginal and contemplated by many researchers, remained unimplemented due to resource constraints, elevated API costs, and the inaccessibility of open-source LLMs such as PaLM.

2. Moreover, the proposition that the approach mimics human learning—by considering multiple answers, determining the solution, and then learning or committing the solution to memory—is unsupported and seemingly arbitrary. This assertion equates human problem-solving with repeatedly contemplating solutions, concluding the most probable one, and then using it for learning or memorization. This point is highly contentious and warrants further clarification from the authors.

3. Additionally, research asserting such a significant claim—that LLMs have the capacity for self-improvement—must demonstrate evidence across a range of models, from small to extremely large. Solely exhibiting results on a 540B model does not substantiate the claim that large LLMs can enhance themselves by training on their produced samples. For instance, when utilizing smaller models like T5-large, the accuracy may be insufficient, and the generated answers are often incorrect, suggesting that this methodology may only be applicable to PaLM or models of similar size.

4. Lastly, the experimentation's confinement to a single, non-open-source model does not sufficiently support the overarching assertion that "large language models can self-improve." The lack of reproducibility in the results raises concerns for many within the research community. The method's applicability is limited to the 540B model, which is not widely accessible or feasible for most researchers to train or fine-tune. This raises questions about the method's effectiveness when applied to smaller LLMs (e.g. LLaMA, etc.) that are more widely available. It would be beneficial to investigate whether these approaches can be adapted for language models that are accessible to the broader community and operate at more manageable scales. Replicating the results of this method is extremely challenging, making it nearly impractical for other researchers to validate or build upon the findings.

**Reproducibility:**

2: Would be hard pressed to reproduce the results. The contribution depends on data that are simply not available outside the author's institution or consortium; not enough details are provided.

**Reviewer Confidence:**

5: Positive that my evaluation is correct. I read the paper very carefully and I am very familiar with related work.

---

> ### Author Rebuttal · Authors · 2023-08-29
>
> Thanks for your thoughtful comments! We address your concerns as below:
>
> **Q1: The concept of training on confident samples is unoriginal and remains unimplemented due to resource constraints?**
>
> A1: In addition to using self-generated diverse paths as training data, we design a set of mixed formats to further augment the training data (in Table 2 and Sec 3.2). During training, the model tries to use inferior prompts (w/o chain-of-thought, zero-shot, etc.) to achieve results that it could only achieve with few-shot chain-of-thought prompting, and this let the model learn to reason on its own and to reason on out-of-domain questions. We find this to be empirically helpful for improving the language model reasoning ability as shown in the table below. We will include this table in the next version of the paper if space allows.
>
> | Training Formats | 	Std Prompting | CoT Prompting |
> | ---- | ---- | ---- |
> | w/o LMSI | 17.9 | 56.5 |
> | LMSI w. Format 1 in Table 2 | 29.2| 69.4 |
> | LMSI w. All Formats | 32.2 | 73.5 |
>
> We also propose novel techniques for self-improvement for two extreme low-resource scenarios in Sec 3.3: (1) question generation where only a few example questions are provided and (2) prompt generation for zero-shot learning setting.
>
> **Q2: The proposition of mimicking human learning is unsupported?**
>
> A2: Our analogy to human learning is not rigorous, and we will remove this analogy in the revision.
>
> **Q3: The method must be demonstrated across models from small to extremely large?**
>
> A3: In our paper, we do include experiment results on a smaller model UL2 (20B) in Appendix A.1.
> We further conduct experiments on the LLaMA model. We show the results on 13B and 65B LLaMA models on the GSM8K dataset as below.
>
> | Model | | CoT Prompting | Self Consistency |
> | ---- | ---- | ---- | ---- |
> | LLaMA 13B |  |  |  |
> | &emsp; w/o LMSI |  | 17.8 | 29.3 |
> | &emsp; w. LMSI |  | 25.4 | 32.9 |
> | LLaMA 65B |  |  |  |
> | &emsp; w/o LMSI |  | 50.9 | 69.7 |
> | &emsp; w. LMSI |  | 66.8 | 73.5 |
>
> According to our new experiment results, smaller models can also be improved by LMSI, and the improvement increases with model size. We will include results of remaining datasets in the next version of our paper if space permits.
>
> We further explain the reason why self-improvement increases with the size of language model: A recent study points out that language model calibration [1] increases with model size. Therefore, the larger the model size, the more accurate the most confident reasoning paths, and the higher quality of generated training data leads to better self-improvement.
>
> **Q4: Is the method effective on accessible language models such as LLaMA or other models?**
>
> A4: Yes, please find the results of LMSI on UL2 (20B) in Appendix A.1 and results on LLaMA models (13B, 65B) in Q3.
>
>
> [1] Language Models (Mostly) Know What They Know. https://arxiv.org/pdf/2207.05221.pdf

---

### Official Review · Reviewer_AVsp · 2023-08-04

**Soundness:** 4

**Excitement:**

4: Strong: This paper deepens the understanding of some phenomenon or lowers the barriers to an existing research direction.

**Missing References:**

* In Sec 3.3 "Prompt Generation", this work may be relevant: https://arxiv.org/abs/2210.03493
* In Sec 5.2, Line 478, maybe another potential reason is the benefit of using CoT format: https://arxiv.org/abs/2212.10001
* Typo in the citation for ANLI dataset? https://arxiv.org/abs/1910.14599

**Paper Topic And Main Contributions:**

This paper proposes language model self-improved (LMSI), a method to self-train large language models (LLMs) to do complex language tasks with minimal annotation. In LMSI, a model is given a chain-of-thought prompt and an unlabeled dataset, which enables to model to generate diverse reasoning paths on the dataset. Then, reasoning paths with high-confidence are selected, processed into multiple formats, and used to further train the LLM.
Empirical results suggest that LMSI can significantly improve performance on multiple complex tasks. For example, on GSM8K, performance is improved from 74.4% to 82.1%. Additionally, LMSI is also useful (1) for improving out-of-domain task performance; (2) in extreme cases where the question-only dataset is generated or the chain-of-thought prompt is generated.

**Questions For The Authors:**

* In figure 2, is the accuracy computed based on ground truth data? Is this trend consistent across tasks? Raising this question because LSMI are applied to "question-only" dataset without annotation, so there may be risks when applying LSMI to a tasks without this trend.
* Line 425, what are the considerations when choosing these OOD tasks? Also, does "w/o LMSI" in Table 4 mean using the vanilla LLM or using the LLM multi-task fine-tuned on the in-domain tasks? Comparing with the latter would make a more convincing case for LMSI.

**Reasons To Accept:**

* Extensive empirical results on various datasets (GSM8K, DROP, ARC, etc.) and different settings (e.g., multi-task training on in-domain tasks and testing on out-of-domain tasks; generating the questions, generating the CoT prompt), supporting the effectiveness of LMSI.
* Carefully compared and identified key design choices (e.g., mix format) in LMSI.
* A very detailed and well-written related work section which summarizes the literature well and helps the reader to understand the background of the proposed work.


**Reasons To Reject:**

I think the paper is in general very exciting! My main concern is with the title "Large Language Models Can Self-Improve".
  * The title seems to be implying smaller models cannot, but this is unclear for now as it is not systematically evaluated (there was one set of experiment on UL2). Past works show that smaller LMs can "self-improve" too, perhaps in a slightly different setup (e.g., iPET method in https://arxiv.org/abs/2001.07676; https://aclanthology.org/2021.naacl-main.185/).
 * I believe what's unique in this work is the CoT and self-consistency capabilities in the 540B LLMs (also mentioned by the author in line 189-190), which enables a better starting model, and also synergies with self-training technique better, leading to the largest improvement in the end. Personally I think "chain-of-thought enables better self-improvement in LLMs" may be a more appropriate summary.

Some minor concerns are in the sections below.

**Reproducibility:**

3: Could reproduce the results with some difficulty. The settings of parameters are underspecified or subjectively determined; the training/evaluation data are not widely available.

**Reviewer Confidence:**

4: Quite sure. I tried to check the important points carefully. It's unlikely, though conceivable, that I missed something that should affect my ratings.

**Typos Grammar Style And Presentation Improvements:**

* I'm a little confused at the claim on FLAN/T0 requiring high-quality supervised dataset (Line 48-51). I understand Table 4 contains experiments in a setting similar to FLAN/T0 setting and supports this claim, but maybe more context is needed in line 48-51.

---

> ### Author Rebuttal · Authors · 2023-08-29
>
> Thanks for your positive feedback and constructive comments! We address your concerns as below.
>
> **Q1: The title indicates smaller models cannot self-improve?**
>
> A1: Thanks for the great suggestions! We will include the discussions of these two papers in our revision, and make clarifications at the beginning of our paper by pointing out that there are various setups for improving language models. In addition, we also conduct experiments on smaller LLaMA models. We show the results on 13B and 65B LLaMA models on the GSM8K dataset as below.
>
> | Model | | CoT Prompting | Self Consistency |
> | ---- | ---- | ---- | ---- |
> | LLaMA 13B |  |  |  |
> | &emsp; w/o LMSI |  | 17.8 | 29.3 |
> | &emsp; w. LMSI |  | 25.4 | 32.9 |
> | LLaMA 65B |  |  |  |
> | &emsp; w/o LMSI |  | 50.9 | 69.7 |
> | &emsp; w. LMSI |  | 66.8 | 73.5 |
>
> We will include results of remaining datasets in the next version of our paper if space permits.
>
> **Q2: “Chain-of-thought enables better self-improvement in LLMs” is a more appropriate summary?**
>
> A2: Thanks for your advice! We agree that this is a more appropriate summary of the paper, since we empirically show that using chain-of-thoughts instead of direct answer for self-training leads to better improvement. We will consider using a more accurate title in our revision such as your proposed title.
>
> **Q3: Figure 2 computed based on ground truth data? Is this trend consistent across tasks?**
>
> A3: Yes, the accuracy is computed based on ground truth answers. Figure 2 shows the relation between confidence and accuracy of language models, and reflects the calibration of language models. Figure 4 in a recent paper [1] shows that calibration of language models increases with the model size, and their experiments are based on Big Bench [2], a collection of benchmarks from various domains.
>
> **Q4: Considerations of choosing OOD tasks? Meaning of “w/o LMSI” in Table 4?**
>
> A4: We select tasks that have different question styles from the in-domain tasks. For example, the in-domain OpenBookQA and ARC-challenge are commonsense reasoning tasks with 4 choices, while the out-of-domain StrategyQA is a yes-no reasoning task to ask language models whether or not a hypothetical statement is feasible or not.
>
> “w/o LMSI” in Table 4 means using the vanilla LLM. We did not compare with the latter (LLM multi-task fine-tuned on in-domain tasks) because our method is an unsupervised method, and it would be unfair to compare with LLM fine-tuned on the ground truths of in-domain tasks.
>
> **Q5: Missing references for Sec 3.3 and Sec 5.2? Typo of the ANLI citation?**
>
> A5: Thanks for pointing out these relevant studies! We believe that the paper “Automatic Chain of Thought Prompting in Large Language Models” and the paper “Towards Understanding Chain-of-Thought Prompting: An Empirical Study of What Matters” are highly relevant to our work, and we will include them in our next version of the paper. We are sorry for the error in the ANLI citation, and we will correct it in our revision.
>
> **Q6: Flan/T0 needs high-quality supervised dataset?**
>
> A6: Yes, both FLAN [3] and T0 [4] are instruction-tuned models, where they rely on large collections of human curated question and answer pairs [5] for supervised fine-tuning. They focus on constructing diverse **input questions** for training. Our method focuses on generating diverse **output paths** for language model self-training. Our method is completely unsupervised, so the experiments in Table 4 have a different setting with FLAN/T0 (they do have supervisions). Sorry for the confusion and we will clarify this in our revision.
>
>
> [1] Language Models (Mostly) Know What They Know. https://arxiv.org/pdf/2207.05221.pdf
>
> [2] Beyond the Imitation Game: Quantifying and extrapolating the capabilities of language models. https://arxiv.org/pdf/2206.04615.pdf
>
> [3] Scaling Instruction-Finetuned Language Models. https://arxiv.org/pdf/2210.11416.pdf
>
> [4] Multitask Prompted Training Enables Zero-Shot Task Generalization. ICLR 2022.
>
> [5] The Flan Collection: Designing Data and Methods for Effective Instruction Tuning. ICML 2023.

---

### Official Review · Reviewer_p95e · 2023-08-09

**Soundness:** 4

**Excitement:**

4: Strong: This paper deepens the understanding of some phenomenon or lowers the barriers to an existing research direction.

**Paper Topic And Main Contributions:**

The authors employ a CoT approach to prompt diverse reasoning pathways in large language models. They utilize self-consistency to filter high-confidence answers, generating a self-supervised dataset. Based on this dataset, a novel fine-tuning strategy is proposed to enhance the model's reasoning abilities. Substantial performance gains are demonstrated on tasks such as GSM8K and openBookQA.

**Questions For The Authors:**

A. What motivated the choice of the 540B model? Have experiments been conducted on slightly smaller models? Can the conclusions drawn from this method be generally applied?

B. Could additional tuning of the model negatively impact its performance on tasks like MMLU, potentially leading to catastrophic forgetting?

C. How does the LMSI method relate to instruction-based tuning? Are these approaches complementary?

**Reasons To Accept:**

+ The approach of constructing a dataset through self-generated model outputs is innovative.
+ The method is straightforward yet effective, yielding improvements across multiple reasoning tasks.
+ The effectiveness of the proposed method is validated across various scenarios, including resource-scarce tasks.

**Reasons To Reject:**

- The potential negative impact of further tuning the LLM, on non-reasoning tasks has not been empirically verified in the paper.
- The validation of this method has been exclusively conducted on the massive 540B model, lacking generalization studies across models of smaller scales such as 7/13/70B.

**Reproducibility:**

3: Could reproduce the results with some difficulty. The settings of parameters are underspecified or subjectively determined; the training/evaluation data are not widely available.

**Reviewer Confidence:**

4: Quite sure. I tried to check the important points carefully. It's unlikely, though conceivable, that I missed something that should affect my ratings.

---

> ### Author Rebuttal · Authors · 2023-08-29
>
> Thanks for your valuable feedback! We address your concerns as below.
>
> **Q1: Potential negative impact of further tuning LLM on non-reasoning tasks?**
>
> A1: This is a well acknowledged observation of current LLMs [1]. A recent paper [2] also figures out that continual training of the GPT-3.5 and GPT-4 models lead to performance downgrade in some tasks. We believe this is currently still an open problem and needs further explorations.
>
> **Q2: Studies on smaller models?**
>
> A2: In our paper, we do include experiment results on a smaller model UL2 (20B) in Appendix A.1.
> We further conduct experiments on the LLaMA model. We show the results on 13B and 65B LLaMA models on the GSM8K dataset as below.
>
> | Model | | CoT Prompting | Self Consistency |
> | ---- | ---- | ---- | ---- |
> | LLaMA 13B |  |  |  |
> | &emsp; w/o LMSI |  | 17.8 | 29.3 |
> | &emsp; w. LMSI |  | 25.4 | 32.9 |
> | LLaMA 65B |  |  |  |
> | &emsp; w/o LMSI |  | 50.9 | 69.7 |
> | &emsp; w. LMSI |  | 66.8 | 73.5 |
>
> According to our new experiment results, smaller models can also be improved by LMSI, and the improvement increases with model size. We will include results of remaining datasets in the next version of our paper if space permits.
>
> **Q3: Choice of the 540B model? Can the conclusions drawn from this method be generally applied?**
>
> A3: Please find more results on smaller models (13B and 65B) in Q2. We observe that self-improvement increases with model size. We claim that this is because larger models are more well-calibrated [3], which means that the highest confident answers are more likely to be accurate. Therefore, the larger the model, the more accurate the most confident reasoning paths, and the higher quality of generated training data leads to better self-improvement.
>
> **Q4: Additional tuning of the model negatively impacts its performance on tasks like MMLU, leading to catastrophic forgetting?**
>
> A4: It is a common phenomenon [1] that fine-tuning language models will distort their pre-trained features, and could lead to performance downgrades in out-of-domain tasks. A recent paper [2] also figures out that continual training of the GPT-3.5 and GPT-4 models lead to performance downgrade in some tasks. We mention in Q5 that LMSI could be integrated with instruction-tuning to improve the performance on out-of-domain tasks.
>
> **Q5: relation of LMSI to instruction-based tuning? Are these approaches complementary?**
>
> A5: LMSI and instruction-based tuning improves language models in separate directions: (1) Instruction tuning provides a diverse collection of tasks with different instructions for fine-tuning the model, which improves the zero-shot performance of LLMs on unseen tasks. Their training datasets are often human-annotated [4] or self-generated [5]. This direction improves language models by various **input questions** during training. (2) Our LMSI method trains language models with various **output reasoning paths** as pseudo “labels”, and is completely unsupervised. We believe the two directions could be complementary with each other: For example, we could let the LLM generate multiple decoding traces for each question in the instruction-tuning dataset for self-improving, which could then save the amounts of data needed for instruction-tuning.
>
> [1] Fine-Tuning can Distort Pretrained Features and Underperform Out-of-Distribution. ICLR 2022.
>
> [2] How is ChatGPT’s Behavior Changing over Time? https://arxiv.org/pdf/2307.09009.pdf
>
> [3] Language Models (Mostly) Know What They Know. https://arxiv.org/pdf/2207.05221.pdf
>
> [4] The Flan Collection: Designing Data and Methods for Effective Instruction Tuning. ICML 2023.
>
> [5] Self-Instruct: Aligning Language Models with Self-Generated Instructions. ACL 2023.

---

### Official Review · Reviewer_jR8L · 2023-08-09

**Soundness:** 3

**Excitement:**

4: Strong: This paper deepens the understanding of some phenomenon or lowers the barriers to an existing research direction.

**Paper Topic And Main Contributions:**

The paper addresses the problem of training Large Language Models in the absence of labeled training data, which may be costly to obtain.
To overcome this, the authors outline approaches to generating both pseudo-labels accompanied by reasoning chains *given a question*, as well as generating new training questions themselves.
These are then used to fine-tune Large Language Models (almost all main results are done using a 540B parameter Transformer) on this data, which are then compared to zero-shot prompting them, both with different kinds of prompting mechanisms, and including further ablation studies, for example on the number of reasoning chains that are provided.
The method itself is a form of self-training and applies many existing techniques, such as Chain-of-Thought prompting, or self-consistency sampling, for it.
Furthermore, the data is used for distillation into smaller models, under the acknowledgement of previous related works that have used similar methods (nevertheless, related works have not used pseudo-labels but ground-truth labels, but have created reasoning chains).

**Questions For The Authors:**

(A) In Figure 2, the caption says that "Predicted confidence from self-consistency (Wang et al., 2022b) is well calibrated (Guo et al., 2017)." However, while the plot shows a clear correlation between accuracy and confidence, it is also clearly not well-calibrated in the sense of the cited paper (Guo et al., 2017), as for that to hold the observations would have to lie on the diagonal from 0,0 to 1,1. It would be good if this could be clarified.

(B) Especially with L484-486 in mind, have you checked whether questions from the test set might have been generated?

(C) Which negative results did you find? I think that a discussion of those would also be beneficial for readers.

**Reasons To Accept:**

- The paper is clearly written and structured
- The method presented in the paper is intuitive and design choices are well-motivated
- The improvements gained with the method are substantial and of interest to the community

**Reasons To Reject:**

- The paper mainly consists of an application of existing techniques to a form of self-training, which is itself explored in many prior works (cited by the authors)
- The method is hard to reproduce, due to the reliance on a non-public very large language model.
- While the authors acknowledge the above limitation, they only provide one experiment with a smaller (20B) openly-available model that is only found in the Appendix. I believe that a greater discussion in the main paper would be of benefit.
- The framing of the paper appears overstated to me. in particular seems the notion of self-improvement to mainly be a rewording of self-training. Furthermore, the analogy to humans learning without external inputs in L4-6 is lacking, as even for the generated training questions, there are 10 existing ones created by humans that are used in the prompt (L453).
- It would have been nice to see more comparisons to training on human-labeled data, as this would also give insights on how much human data is actually required to collect to meet the performance, which the authors claim to be a large amount.

**Reproducibility:**

2: Would be hard pressed to reproduce the results. The contribution depends on data that are simply not available outside the author's institution or consortium; not enough details are provided.

**Reviewer Confidence:**

4: Quite sure. I tried to check the important points carefully. It's unlikely, though conceivable, that I missed something that should affect my ratings.

**Typos Grammar Style And Presentation Improvements:**

(A) L11: Chain-of-Though -> Chain-of-Thought
(B) L104: improvments -> improvements
(C) L207, L653-L655: author name contains erroneous white space in citation
(D) L399: increase -> increases
(E) Figure 2: As the figure is essentially a reliability diagram, it would be good to indicate perfect calibration, for example by a dashed line from 0,0 to 1,1. Also, the distances (in mm when plotted) between the units of x and y axis seem to be different, which distorts the plot and obscures conclusions that may be taken by readers.

---

> ### Author Rebuttal · Authors · 2023-08-29
>
> Thanks for your insightful comments! We address your concerns as below.
>
> **Q1: The paper mainly consists of existing techniques (chain-of-thought prompting, self-consistency sampling)?**
>
> A1: In addition to using self-generated diverse paths as training data, we design a set of mixed formats to further augment the training data (in Table 2 and Sec 3.2). During training, the model tries to use inferior prompts (w/o chain-of-thought, zero-shot, etc.) to achieve results that it could only achieve with few-shot chain-of-thought prompting, and this let the model learn to reason on its own and to reason on out-of-domain questions. We find this to be empirically helpful for improving the language model reasoning ability as shown in the table below. We will include this table in the next version of the paper if space allows.
>
> | Training Formats | 	Std Prompting | CoT Prompting |
> | ---- | ---- | ---- |
> | w/o LMSI | 17.9 | 56.5 |
> | LMSI w. Format 1 in Table 2 | 29.2| 69.4 |
> | LMSI w. All Formats | 32.2 | 73.5 |
>
> We also propose novel techniques for self-improvement for two extreme low-resource scenarios in Sec 3.3: (1) question generation where only a few example questions are provided and (2) prompt generation for zero-shot learning setting.
>
> **Q2: Hard to reproduce?**
>
> A2: We conduct additional experiments on the LLaMA model. Please refer to Q3.
>
> **Q3: Greater discussion of smaller models?**
>
> A3: In addition to the UL2 (20B) model in our paper, we further conduct experiments on the LLaMA model. We show the results on 13B and 65B LLaMA models on the GSM8K dataset as below.
>
> | Model | | CoT Prompting | Self Consistency |
> | ---- | ---- | ---- | ---- |
> | LLaMA 13B |  |  |  |
> | &emsp; w/o LMSI |  | 17.8 | 29.3 |
> | &emsp; w. LMSI |  | 25.4 | 32.9 |
> | LLaMA 65B |  |  |  |
> | &emsp; w/o LMSI |  | 50.9 | 69.7 |
> | &emsp; w. LMSI |  | 66.8 | 73.5 |
>
> According to our new experiment results, smaller models can also be improved by LMSI, and the improvement increases with model size. We will include results of remaining datasets in the next version of our paper if space permits.
>
> **Q4: Framing overstated? Lack analogy to human learning?**
>
> A4: Our analogy to human learning is not rigorous, and we will remove this analogy in the revision. For our framing of the work, we will consider using a more accurate title, such as “Self-improving LLMs with Chain-of-Thoughts'', to emphasize that using chain-of-thoughts instead of direct answers for self-training leads to better improvement.
>
> **Q5: Comparable to how much human-labeled data?**
>
> A5: In our work we did not study the performance of Supervised Fine-Tuning (SFT), but we found a very recent work [1] that compares In-Context Learning (ICL) with different amounts of supervised data on the GSM8K dataset. They found that SFT performance improves slower than ICL as the model size increases. We believe this paper could be relevant to your question.
>
> **Q6: Clarification of well calibration?**
>
> A6: Thanks for pointing this out! We are sorry for this confusing sentence in the paper. According to Figure 4 in this recent paper [2], language model calibration increases with size. In our revision, we will revise the plot by adding a diagonal from (0,0) to (1,1) and clarify that current language models are not perfectly-calibrated though their calibration increases with model size.
>
> **Q7: Whether questions from the test set might have been generated?**
>
> A7: The two approaches in Sec 3.3 are separate parts: the prompt generation approach does not use any generated questions or training questions. The question generation approach uses 10 training questions as seeds, and there are no overlaps between the generated questions and test questions.
>
> **Q8: Any negative results?**
>
> A8: We expected filtering the self-generated reasoning paths and only choosing the highly confident ones (e.g., confidence > 0.6) would benefit model self-training. However, we empirically figured out this did not work better than using all the self-generated paths. We think this is because though the reasoning paths are more confident, the number of training data becomes lower and the model cannot learn from diverse reasoning paths.
>
> **Q9: Typos and presentation improvements?**
>
> A9: Thanks for pointing out these errors and potential improvements! We will fix all of them in our revision.
>
> [1] Scaling Relationship on Learning Mathematical Reasoning with Large Language Models. https://arxiv.org/pdf/2308.01825.pdf
>
> [2] Language Models (Mostly) Know What They Know. https://arxiv.org/pdf/2207.05221.pdf

---

### Meta-Review · Area_Chair_Q3Ze · 2023-09-19

**Recommendation:** 5

**Metareview:**

The paper addresses the challenge of training LLMs when obtaining labeled training data is costly or unavailable. The method presented is both intuitive and supported by well-motivated design choices, enhancing its accessibility and promise. Importantly, the method yields substantial improvements in various reasoning tasks, demonstrating its relevance to the research community. Additionally, the paper is well-written and structured. The authors have addressed the concerns raised by the reviewers. Therefore, I recommend accepting the paper.

---

### Decision · Program_Chairs · 2023-10-07

**Decision:**

Accept-Main

**Comment:**

The paper addresses the challenge of training LLMs when obtaining labeled training data is costly or unavailable. The method presented is both intuitive and supported by well-motivated design choices, enhancing its accessibility and promise. Importantly, the method yields substantial improvements in various reasoning tasks, demonstrating its relevance to the research community. Additionally, the paper is well-written and structured. The authors have addressed the concerns raised by the reviewers. Therefore, I recommend accepting the paper.